# Data science approaches provide a roadmap to understanding the role of abscisic acid in defence

## Original Research Article

ABA; decision tree; machine learning; plant hormone; resistance.

**Authors for correspondence:** K. Stevens, E. Luna, E-mail: kxs1006@student.bham.ac.uk; e.lunadiez@bham.ac.uk

Katie Stevens[1], Iain G. Johnston[2,3] and Estrella Luna[1]

[1]School of Biosciences, University of Birmingham, Birmingham, United Kingdom; [2]Department of Mathematics, University of Bergen, Bergen, Norway; [3]Computational Biology Unit, University of Bergen, Bergen, Norway

## Abstract

Abscisic acid (ABA) is a plant hormone well known to regulate abiotic stress responses. ABA is also recognised for its role in biotic defence, but there is currently a lack of consensus on whether it plays a positive or negative role. Here, we used supervised machine learning to analyse experimental observations on the defensive role of ABA to identify the most influential factors determining disease phenotypes. ABA concentration, plant age and pathogen lifestyle were identified as important modulators of defence behaviour in our computational predictions. We explored these predictions with new experiments in tomato, demonstrating that phenotypes after ABA treatment were indeed highly dependent on plant age and pathogen lifestyle. Integration of these new results into the statistical analysis refined the quantitative model of ABA influence, suggesting a framework for proposing and exploiting further research to make more progress on this complex question. Our approach provides a unifying road map to guide future studies involving the role of ABA in defence.

## 1. Introduction

Abiotic and biotic defence responses are important evolutionary mechanisms that allow plants to adapt to their dynamic environments (Peck & Mittler, 2020). A variety of sophisticated signalling networks modulate plant responses to challenges, helping to enhance survival. Hormone signalling and crosstalk play major roles in these responses, which can also be influenced by plant age, organ type and environment (Berens et al., 2017). However, as plants encounter many biotic challenges, including pathogens with varying lifestyles and methods of attack, there is no one-size-fits-all approach to defence. Therefore, hormones have unique roles in multiple, often antagonistic, complex defence pathways (Robert-Seilaniantz et al., 2011).

Abscisic acid (ABA) is an essential hormone involved in both biotic and abiotic stress signalling (Lee & Luan, 2012). ABA is found universally across vascular plants and is involved in a wide range of physiological processes. For instance, through interaction with developmental hormones, ABA modulates plant growth, seed germination, stomata closure and development and embryo maturation (Vishwakarma et al., 2017). Moreover, ABA is essential for adaptation to drought and salt stress (Casson & Hetherington, 2010; Sah et al., 2016) and has a positive role in ensuring plant survival during abiotic stress. For example, water supply limitation is associated with an increase in ABA across the entire plant that leads to reduced stomatal conductance (Vishwakarma et al., 2017). However, the function of ABA in defence against biotic challenges, including pathogens, is varied, which has led to mounting controversy in the field of plant biology (Asselbergh et al., 2008a; Cao et al., 2011; Ton et al., 2009).

ABA is widely considered to be a positive influence against early pathogen invasion due to its essential role in stomatal closure (Melotto et al., 2006). However, controversy arises in the later infection process. On one hand, ABA is known to be involved in multiple defence pathways and often interacts antagonistically with other hormones such as salicylic acid (SA), jasmonic acid (JA) and ethylene (ET) through 'hormone crosstalk' (Anderson et al., 2004; De Torres Zabala et al., 2009; Oide et al., 2013). SA is involved in resistance against biotrophs and its production is enhanced following colonisation with the bacterial biotroph *Pseudomonas syringae* (Delaney et al., 1995). However, SA has a negative role in resistance to necrotrophs

such as *Botrytis cinerea* (El Oirdi et al., 2011). In contrast, JA and ET display the reverse effect of SA and enhance resistance against necrotrophs and herbivory (Mcdowell & Dangl, 2000). These hormones therefore behave antagonistically, and ABA is typically shown to have a negative influence on resistance as it can disrupt SA-, JA- and ET-mediated defence signalling (Anderson et al., 2004; Audenaert et al., 2002). On the other hand, ABA has been linked to the expression of defence responses controlled by these plant hormones, including deposition of callose and reactive oxygen species production, which mediate resistance against biotrophic and necrotrophic pathogens (Asselbergh et al., 2007). In addition, crucially, a positive role of ABA in defence has been linked to the requirement of an intact ABA signalling pathway for direct and priming of callose deposition (Flors et al., 2005; 2008; Luna et al., 2011; Schwarzenbacher et al., 2020; Ton & Mauch-Mani, 2004; Vicedo et al., 2009). Therefore, the role of ABA in the expression of defence mechanisms appears highly varied, with positive and negative effects via different mechanisms during different circumstances.

The effect of ABA has not previously appeared to be associated with plant species or pathogen lifestyle (Asselbergh et al., 2008b), and contradictions have been shown even within a single model plant pathosystem. For example, whereas ABA-treated Arabidopsis plants display enhanced susceptibility against *P. syringae* (Melotto et al., 2006, Mohr & Cahill, 2007), ABA-deficient *aba3–1* Arabidopsis mutant displays enhanced susceptibility against *P. syringae* (Melotto et al., 2006). In economically important crops such as tomatoes, ABA deficient *sitiens* tomato mutants have enhanced resistance against *B. cinerea*, *Odium neolycopersici* and *Erwinia chrysanthemi* (Achuo et al., 2006; Asselbergh et al., 2007; 2008b; Audenaert et al., 2002). In contrast, long-lasting induced resistance in tomato fruit is marked by an accumulation of ABA (Wilkinson et al., 2018). ABA is therefore a key component of defence at all phases of infection, yet its influence remains to be understood.

To make progress in our understanding of such complex questions, machine learning (ML) and supervised learning techniques are increasingly being used for studying plant pathogen interactions, specially motivated by the increased production of large-omics datasets and imaging capabilities (Sperschneider, 2020; Sun et al., 2020; van Dijk et al., 2021). For instance, ML has facilitated analysis of gene networks in immune response (Dong et al., 2015), pathogen effector protein localisation (Sperschneider et al., 2018) and plant disease diagnostics using image-based learning (Mishra et al., 2019). Here, we have used ML strategies to investigate the contrasting effects of ABA in resistance against pathogens. Through the collection of hundreds of data points from a total of 30 scientific peer-reviewed publications, we used tools from ML to attempt to unpick the complex interplay of effects shaping heterogeneous ABA influences, combined with new experiments to further explore these data-driven findings. Overall, our study highlights key factors that influence the effect of ABA in resistance and provides a tool for future studies towards facilitating the understanding of the role of ABA in plant responses against biotic stresses.

## 2. Methods

### 2.1. Data collection and preparation

Quantitative data regarding disease resistance were collected during March–June 2020 from literature involving ABA (Supplementary Table S1). We searched Web of Science using broad search terms including 'abscisic acid', 'plant pathogen' and 'disease resistance'. Papers were selected if they included quantitative data regarding disease susceptibility from experiments directly involving pathogen infection and manipulation of ABA either through the use of ABA signalling and biosynthesis mutants, exogenous application of ABA, and/or ABA inhibitors. From those papers, 13 shared variables were recorded from the three general classes of plant, ABA treatment, and pathogen. From the plant, we recorded the species, variety, genotype, mutant, mutant type, plant age and tissue. From ABA, we recorded the treatment, application method, and concentration applied. From the pathogen, we recorded the type, lifestyle and species. All variable information was taken directly from papers used; however, values of plant age for orange, grape and pepper were imputed based on known cultivar information.

A large number of disease scoring methods were used across publications such as lesion diameter (mm), % spreading lesions, and categorical scoring. A disease severity index (DSI) was therefore used in an attempt to unify the heterogeneity of disease scoring in our dataset. All observations involving ABA (ABAphenotype) were compared to the level of resistance in control treatments (Controlphenotype) for each individual publication. DSI was represented as a 'percentage resistance change' of all data points, forming a continuous measure of positive and negative percentage resistance changes as a result of ABA, as described below (Supplementary Table S1, Figure 1). For example, the observations of lesion diameter of 5 mm in the Controlphenotype and 8 mm in the ABAphenotype would result a DSI of −60%, thus indicating a susceptible phenotype for our data input.

$$\text{'Percentage resistance change'} =$$
$$100 - (\text{ABAphenotype} * 100/\text{Controlphenotype}) = x\%$$

### 2.2. Computational meta-analysis

Two supervised learning approaches, described below, were used to identify the most important factors to predict disease resistance (Breiman et al., 1984). All analysis was conducted in R (version 4.1.1). For each method of classification, data were randomly split into training (75%) and test (25%) data. Models were generated using the training data and model accuracy was assessed using test data. Final models were generated using all data available. Two models were created: a decision tree (DT) predicting a binary response of resistance 'susceptible' or 'resistant', depending on if the percentage resistance change was a positive or negative value, and a random forest (RF) model. The same variables and parameters were used for both DT and RF. To decide which variables to include in our models and guard against overfitting, the Akaike information criterion (AIC) score was used, providing a trade-off between the maximum likelihood fit of the model on the training data set and the complexity of the model, reflected by the number of parameters. The model with the lowest AIC score was selected.

### 2.3. Classification and regression tree

DTs were constructed using the package rpart (version 4.1–15) (Therneau et al., 2015) with the data described in Supplementary Tables S1 and S2. DT1 describes data collected from existing literature exclusively (Supplementary Table S1), DT2 includes observations from new experiments described in this study in an attempt to refine our original model (Supplementary Table S2). Rpart uses the classification and regression tree (CART)

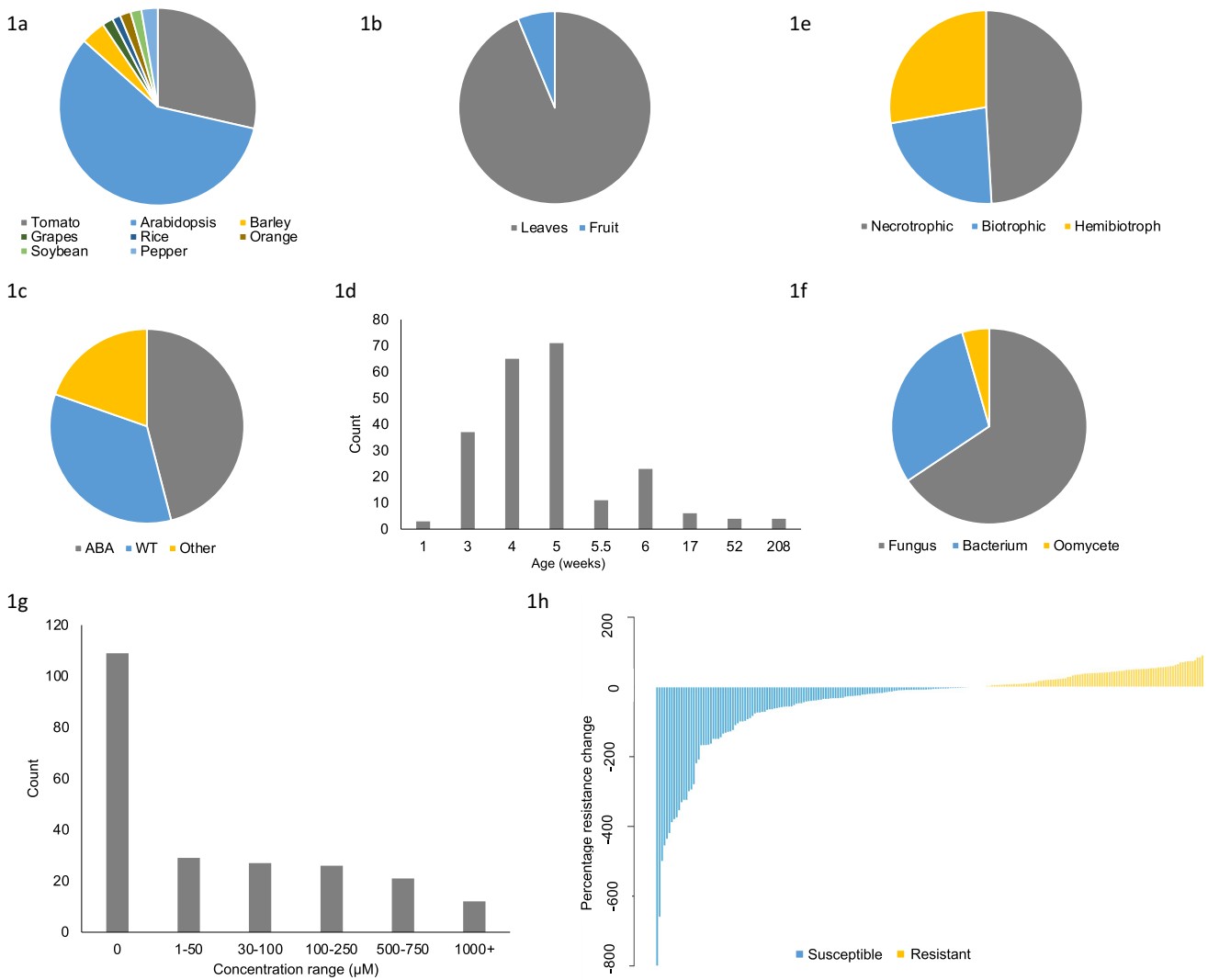

**Fig. 1.** Characteristics of our amalgamated dataset. Proportion of plant species (a), plant tissue (b), mutant type (c), count distribution of plant ages (weeks) (d), pathogen lifestyle (e), pathogen type (f), and count distribution of exogenous ABA concentration range (g). (h) Distribution of percentage resistance value. Data are described in Supplementary Table S1.

algorithm to construct trees based on recursive binary partitioning, which splits the data based on the best predictive variable. The chosen variables and specific splits are determined by the Gini impurity index which measures the disorder in a set of data points. Gini impurity is calculated as the probability of incorrectly labelling a data point assuming random labelling according to the distribution of all classes in the set (Breiman et al., 1984). Splits in trees therefore aim to maximise the decrease in impurity; this process is repeated to build the tree. Seven variables were selected for DT analysis based on the optimal AIC score. Trees produced via the CART algorithm are typically large and overfitted, to reduce complexity trees are pruned. Pruning is determined by the rpart 'complexity parameter' value of the smallest tree with the smallest cross-validation error (Therneau et al., 2015). The CART algorithm additionally provides a measure of variable importance. Variable importance is determined by the accumulated contribution of each variable across all nodes and trees where it is used (Breiman, 2001; Breiman et al., 1984). Precision and recall statistics of all models were calculated from true-positive (TP), false-positive (FP), true-negative (TN) and false-negative (FN) values. The final model predictive performance

was determined by creating receiver-operating characteristic (ROC) plots and obtaining area under curve (AUC) values. ROC plots and AUC values were made using the package ROCR (version 1.0–11) (Sing et al., 2005).

## 2.4. Random forest

RF analysis was performed using the package randomforest (version 4.6–14) (Liaw & Wiener, 2002) using the data described in Supplementary Tables S3 and S4. RF uses the CART algorithm, as described above, to construct a set of different individual trees, which are assembled into a 'forest'. Here, each tree provides a classification 'vote', and the average vote of all trees in the forest is selected. Through resampling with replacement, RFs guard against overfitting, and RFs can improve accuracy and predictive abilities in classification compared to singular CART trees (Breiman, 2001). In our approach, the number of trees and cross-validation was optimised to provide the lowest out-of-bag (OOB) estimate of error rate. The final RF model used in this study contained 400 trees and 10 folds cross-validation.

## 2.5. Simplified machine learning models

Further DT and RF analysis were performed to test the importance of specific variables using the following minimised datasets: DText/RFext–entire dataset minus extreme exogenous concentrations of ABA (i.e., above 750 $\mu$M) and DTage/RFage – entire dataset minus plant age over 6 weeks.

## 2.6. Plant materials and growth conditions

Tomato seeds (cultivar micro-tom) were maintained in damp, humid, dark conditions to stimulate germination. Germinated seeds were transferred to individual 80 mL pots containing Scott's Levington M3 soil. For analysis of older plants, 3-week-old plants were transplanted into 3 l pots until experiments were performed. Plants were grown in greenhouses under a daily light integral of 8.64 mol m$^{-2}$ d$^{-1}$ and 25°C /20°C temperatures. Experiments were performed between July and November 2020.

## 2.7. Abscisic acid treatment

Leaf material was treated with a freshly prepared stock of 20-mM ABA (De Torres Zabala et al.) (Sigma Catalogue number A1049) dissolved in ethanol. The ABA stock was diluted in water to the different concentrations used in this study: 20, 50, 100, and 500 $\mu$M. The control treatment was made by applying water. All treatment solutions were adjusted to the amount of ethanol present, determined by the highest concentration of ABA. Leaves from 4-week and 8-week-old plants were excised and placed in a plastic tray, laid flat, with stems wrapped in wet paper as previously described (Luna et al., 2015; Worrall et al., 2012). Two leaves per plant and a total of six plants (six biological replicates) were used to test the effect of ABA in leaves. ABA treatment was performed by spraying the treatment solutions onto leaves. Leaves were then allowed to dry before moving trays to 100% humidity in the dark for 24 h before pathogen infection.

## 2.8. Botrytis cinerea cultivation and infection

*Botrytis cinerea* (isolate BcI16) was grown on plates of potato dextrose agar (PDA;Difco) and plates were kept in the dark at room temperature for 4 weeks. Infections of leaves with *B. cinerea* were performed as previously described (Luna et al., 2015) by applying droplets of 5 $\mu$L inoculum containing 5x10$^5$ spores/ml. Leaves were incubated in the dark at 100% humidity and 20 °C. Disease severity was recorded at 3 days' post-infection (dpi) by measuring the diameter of the necrotic lesions using a Vernier calliper.

## 2.9. Phytophthora infestans cultivation and infection

*Phytophthora infestans* 88069td10 was grown on rye medium supplemented with geneticin for 2–3 weeks at 20°C as previously described (Whisson et al., 2007). Plates were flooded with 10 mL water and spores scraped from the plate surface and incubated at 4°C for 1–3 hours. Zoospores released from sporangia were decanted. A desired concentration of 5 x 10$^4$ spores/ml was obtained. The underside of each leaf was infected with six droplets of 10 $\mu$L inoculum. Plants were incubated at 100% humidity, 16 hr/8 hr light/dark cycles and 18°C /15°C temperature cycle. Disease severity was recorded at 7 days' post-infection. Scoring was done by classifying lesions into different categories of colonisation. Class I: healthy, Class II; lesions on leaf underside only, Class III; lesions in under and upper leaf surface, Class IV: prominent lesions on both leaf sides, Class V: spreading lesions causing tissue damage, Class VI: total leaf collapse. Category distributions were used to calculate DSIs by using the following formula (Chiang et al., 2017):

$$Disease\ severity\ index = \sum \frac{(Class\ frequency \times score\ of\ rating\ class)}{(Total\ number\ of\ observations \times maximal\ disease\ index)}$$

## 2.10. Statistical analysis

Data from the experiments with *B. cinerea* and *P. infestans* were analysed using R (version 4.1.1). In the analysis of lesion diameter and DSI, normality of distributions was assessed through qq plots and Shapiro–Wilk tests, and homogeneity of variance was assessed through Levene's tests. Data for which the null hypothesis of normality was not rejected were analysed through one-way analysis of variance (ANOVA) data; cases where normality was rejected were analysed with Kruskal–Wallis tests. Distributions with homogenous variances were analysed through least significant differences (LSD) post-hoc tests. Distributions with non-homogeneous variances were analysed with Dunnett's post-hoc tests. When data did not support rejection of normality, two-tailed *t*-tests were used to compare differences between water control and grouped treatments of different ABA concentrations. Experiments were repeated twice with similar results.

## 3. Results

### 3.1. Data exploration and sorting

A total of 224 data points were collected from a total of 30 peer peer-reviewed papers directly reporting resistance phenotypes (Figure 1, Supplementary Table S1). A total of 194 data points were from studies using Arabidopsis or tomato as the experimental species, representing 58% and 29% of the data, respectively (Figure 1a). The majority of data represented experiments conducted in leaves, with only 14/225 data points in fruit tissue (Figure 1b). There were 55 total different genotypes in this dataset that were categorised by type of mutation into three classes: wildtype (WT), ABA mutant (including signalling and synthesis variants), and 'Other' (primarily callose or other hormone-disrupted mutants). ABA mutants represented 46% of the data points (Figure 1c). Regarding plant age, 94% of data points were obtained in plants of the age of or younger than 6 weeks (Figure 1d). Approximately 23%, 49% and 28% of the data points correspond to biotrophic, necrotrophic and hemibiotrophic pathogens, respectively (Figure 1e). Fungi represented the majority of the data with 66% of the points, while bacteria made up 30% and oomycetes made up 4% (Figure 1f). Exogenous concentrations of ABA ranged from 0 to 100,000 $\mu$M, although 95% of the data points were between 10 and 750 $\mu$M (Figure 1g). A total of 109 data points had an exogenous ABA concentration value of 0 due to experiments involving ABA mutants rather than external application. In this dataset, the level of 'percentage resistance change' varied from −800 to 91. A total of 132 data points were labelled 'Susceptible' (negative change in resistance) and 93 data points 'Resistant' (positive change in resistance) (Figure 1h).

### 3.2. CART models

The CART algorithm was used to produce a DT to unpick which variables are involved in, and the most influential in, determining

**Table 1.** Importance of variables in DT and RF determined by rpart and RF

| Model | Variable | Predictive importance |
|---|---|---|
| DT | Exogenous ABA concentration ($\mu$M) | 21.7 |
| | Plant age (weeks) | 13.54 |
| | Plant mutant | 12.98 |
| | Plant species | 11.87 |
| | Pathogen lifestyle | 9.89 |
| RF | Exogenous ABA concentration ($\mu$M) | 35.02 |
| | Plant species | 15.34 |
| | Pathogen lifestyle | 14.55 |
| | Plant age (weeks) | 12.7 |

*Note.* Data described in Supplementary Tables S1 and S3.
Abbreviations: DT, decision tree; RF, random forest.

resistance phenotypes linked to ABA. A DT predicting a binary response of resistance had a predictive accuracy on unseen data of 76% and an AUC value of 77%; the ROC curve is displayed in Supplementary Figure S1. DT had a precision score of 69% and a recall score of 78% (further details in Supplementary Table S6). Rpart determined the most important variables in DT (Table 1). ABA concentration (in $\mu$M) was determined as the most important variable. Plant age (in weeks), plant species and mutant type were also determined as important predictive variables.

### 3.3. Random forest models

An RF model was created for predicting a binary resistance output. Increasing the number of trees used in the forest increased the predictive accuracy of the model; however, beyond 400 trees the predictive capabilities of our model did not improve. The final RF model was therefore generated by using 400 trees. This model had an out-of-bag (OOB) estimate of error rate of 21%. The error rate in unseen data for predicting 'resistance' was 27%, the error rate for predicting 'susceptibility' was 15%, (further details in Supplementary Table S6). As in the DT model, ABA concentration ($\mu$M) was considered the most important variable in the RF model (Table 1). Similar to DT models, RF predicts plant age (in weeks), plant species and pathogen lifestyle as important variables.

### 3.4. Simplified decision trees and random forest analysis

DTs were reconstructed using minimised datasets to assess if accuracy could be improved from 76% and whether alternative predictions would be produced (Supplementary Figure S2; Supplementary Table S5). Removing extreme exogenous concentrations of ABA (i.e., above 750 $\mu$M) created a different DT, which we labelled DText. DText predictive accuracy was reduced to 61% and its AUC value was 77% (further details in Supplementary Table S6). DText has minimal structural changes with a single change in node positioning shifting the plant age threshold from 5.3 to 5.5 weeks (Supplementary Figure S2; Supplementary Table S5). Removing extreme age data points of plants over 6 weeks, creating a different DT labelled DTage, reduced predictive accuracy to 69% and produced a more simplified tree with the root node split of concentration changed to <90 $\mu$M (Supplementary Figure S2; Supplementary Table S5). DTage had a relatively unchanged AUC value of 76% (further details in Supplementary Table S6).

We then tested whether the RF OOB estimate of error could be improved. RFext, constructed by removing extreme concentrations as for DText, marginally increased the OOB error rate to 22% demonstrating that inclusion of these data points in the RF analysis

gives us the best possible prediction within the noise of the data. Finally, the OOB error rate was unchanged in RFage, constructed after removing extreme age points as for DTage (further details for RFext and RFage in Supplementary Table S6).

### 3.5. Decision-tree predictions

The DT (Figure 2) predicted that exogenous ABA concentrations above 38 $\mu$M more frequently lead to susceptibility. This tree predicted that exogenous concentrations below 38 $\mu$M infected with biotrophic pathogens display resistance and those with exogenous ABA concentrations below 38 $\mu$M infected with non-biotrophic pathogens tomato plants display resistance. The DT predicts that exogenous ABA concentrations above 38 $\mu$M infected with non-fungal pathogens display susceptibility. This DT shows the impact of age in resistance: it predicts that with exogenous ABA concentrations above 38 $\mu$M, during fungal infections 'other' mutant plants younger than 5.3 weeks display greater resistance than plants over 5.3 weeks.

### 3.6. Effect of ABA on disease resistance against *Botrytis cinerea* and *Phytophthora infestans*

Our RF and DT results highlighted that exogenous ABA concentration, pathogen lifestyle and plant age are important variables in resistance predictions. Seeking to refine our quantitative understanding of these influences, we therefore experimentally tested these specific variables. Tomato was used as the experimental plant species as it appears as a split in our DT (Figure 2) and is one of the most represented species in our data (Supplementary Table S1).

The effect of exogenous ABA application at different concentrations on susceptibility against *B. cinerea* was assessed in tomato plants of different ages (i.e., 4-week and 8-week-old). ABA treatment had a statistically significant effect on the phenotype in leaves from 4-week-old plants against *B. cinerea* (Figure 3a; ANOVA, $p$ = .04), with ABA generally causing an increase in susceptibility, although post-hoc test (Least Significant Difference - LSD) does not show any specific statistically significant differences. In contrast, in older plants, ABA concentrations of 50 $\mu$M and above enhanced resistance against *B. cinerea* (Figure 3b; Kruskal–Wallis, $p$ = .001). The concentration of 100 $\mu$M led to a significant, dramatic reduction in disease severity (Figure 3b). These results suggest that the effect of ABA on susceptibility against *B. cinerea* is influenced by ABA concentration and the plant age.

The effect of ABA application was also studied against the biotrophic pathogen *P. infestans*. In young leaves, ABA had a statistically significant effect on susceptibility against this pathogen (Figure 3c; ANOVA, $p$ = .012), with the concentration 500 $\mu$M causing increased susceptibility in 4-week-old leaves ($p$ = .029). In older leaves, ABA treatment increased susceptibility in all instances (Figure 3d; ANOVA, $p$ = .04). However, lower range concentrations had a more severe effect on disease incidence, with 20 $\mu$M leading to statistically significant increase in susceptibility ($p$ = .003).

### 3.7. Decision trees with extended experimental data

Comparing our DT predictions (Figure 2) to our experimental results (Figure 3) highlighted similarities and differences in our model. Our DT predicted only 1/8 of our *B. cinerea* infection results correctly. The prediction for *P. infestans* infected plants was more accurate with 6/8 experimental results being predicted by DT correctly. In an attempt to refine an initially imperfect model, data

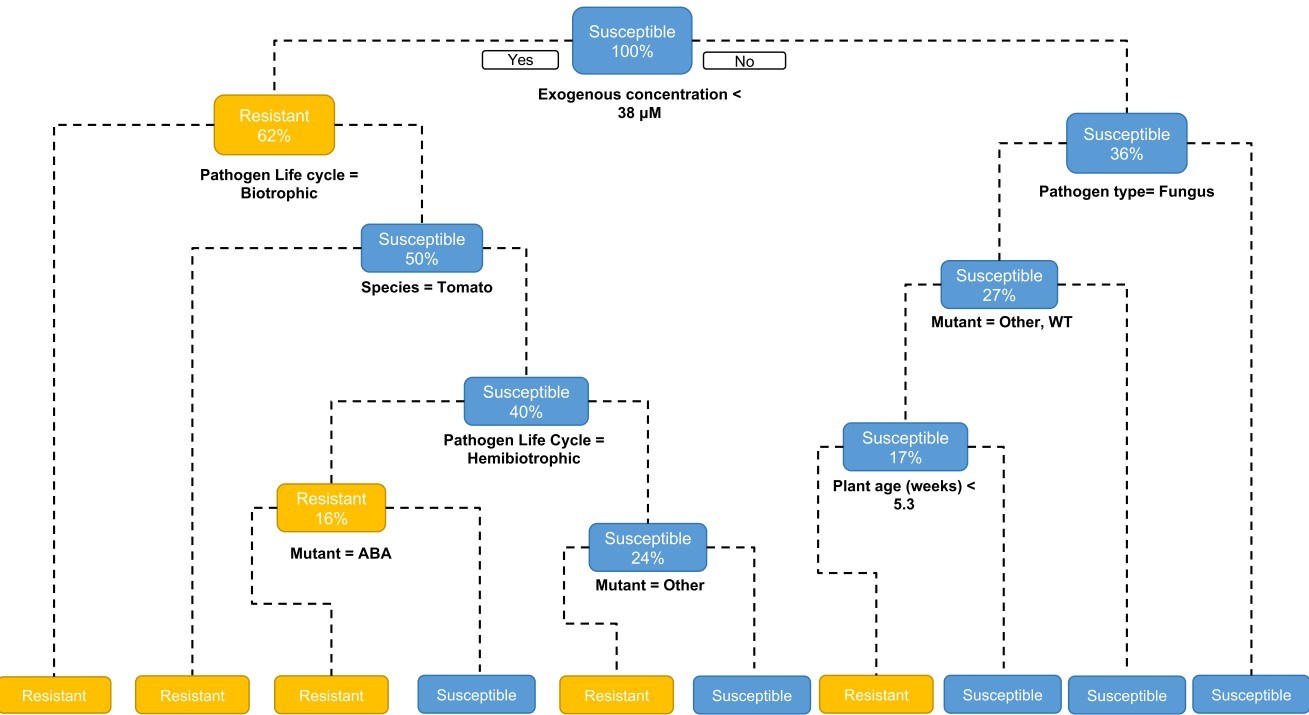

**Fig. 2.** Decision tree (DT) predicting a binary resistance response of 'Susceptible' or 'Resistant'. Yellow boxes indicate resistance and blue boxes indicate susceptibility. Percentage number in box represents proportion of dataset remaining following node split. DT is read from top root node to bottom leaf nodes, if the question asked at a node is 'yes' the left side is followed. Data are described in Supplementary Table S1.

were included from new and rationally chosen experiments (Supplementary Tables S2 and S4). We thus added our experimental findings to our dataset, forming DT2 (Figure 4). The predictive abilities of DT2 remained relatively unchanged, with a predictive accuracy on unseen data of 70%, and an increased AUC value of 85%. DT2 had precision and recall values of 83% and 58%, respectively demonstrating increased precision however lower power (Supplementary Table S6). However, DT2 is now capable of better capturing the new experiments above. For *B. cinerea* infections 6/8 of our experimental results were predicted correctly from DT2, 6/8 were predicted correctly from *P. infestans* infection. The most important variable in DT2 was maintained as exogenous ABA concentration. Addition of our experimental data to our model altered some later splits in the DT2 compared to DT1 leading to differences in the overall topology. The split of age was lowered from 5.3 weeks to 4.5 weeks and an additional node was created. For concentrations higher than 38 $\mu$M a node was created splitting for specific plant species (Arabidopsis, orange, rice, and tomato) and further predicted that exogenous ABA concentrations greater than 375 $\mu$M lead to resistance (Figure 4). These changes subsequently increased the number of final leaf nodes from 10 in DT1 to 11 in DT2, potentially allowing a more fine-grained classification.

## 4. Discussion

The role of ABA in pathogen defence has remained highly ambiguous (Ton et al., 2009). In this study we used supervised ML techniques to attempt to resolve these relationships and determine the most influential factors in ABA resistance phenotypes. In our predictions and experiments, overall, we can observe pronounced roles for ABA on susceptibility to pathogens. Exogenous ABA concentration, pathogen lifestyle and plant age were shown to be important influencers of resistance, therefore illustrating ABA's

regulatory role in the activation of major defence pathways and its specificity in individual pathosystems.

### 4.1. Importance of the concentration

DTs and RF analyses predict the exogenous concentration as the primary determinant factor on the role of ABA. Our DT predicts that exogenous concentrations higher than 38 $\mu$M result in susceptibility (Figure 2). Our experimental work showed in general terms that ABA application at high concentrations is associated with susceptibility (Figure 3). However, our experimental work did not show a consistent pattern of susceptibility as exogenous concentration of ABA was increased or as it passed a particular concentration threshold. For instance, we were unable to observe changes at the threshold concentration predicted and instead variable levels of susceptibility were observed (Figure 3). Moreover, whereas increasing exogenous concentrations of ABA led to enhanced susceptibility, the concentration of 100 $\mu$M was extremely effective in triggering resistance against *B. cinerea* (Figure 3c). The differences observed between the predictions and the experimental work may be in part due to differences in the basal levels of ABA in plants. Achuo *et al.*, demonstrated major discrepancies in the endogenous ABA levels found in Moneymaker tomato plants. Basal quantities varied within identical experimental setups from $4.08 \times 10^{-4}$ $\mu$M (408 picomol) g$^{-1}$ FW to approximately $2.5 \times 10^{-3}$ $\mu$M (2,500 picomol) g$^{-1}$ FW (Achuo et al., 2006). Therefore, basal concentrations of ABA can be different depending on the experimental conditions and could explain discrepancies between predictions and observations.

### 4.2. Importance of the plant age and pathogen lifestyle

DT analysis allowed us to explore the effect of plant age and pathogen lifestyle. Generally, DT predicted that ABA triggers

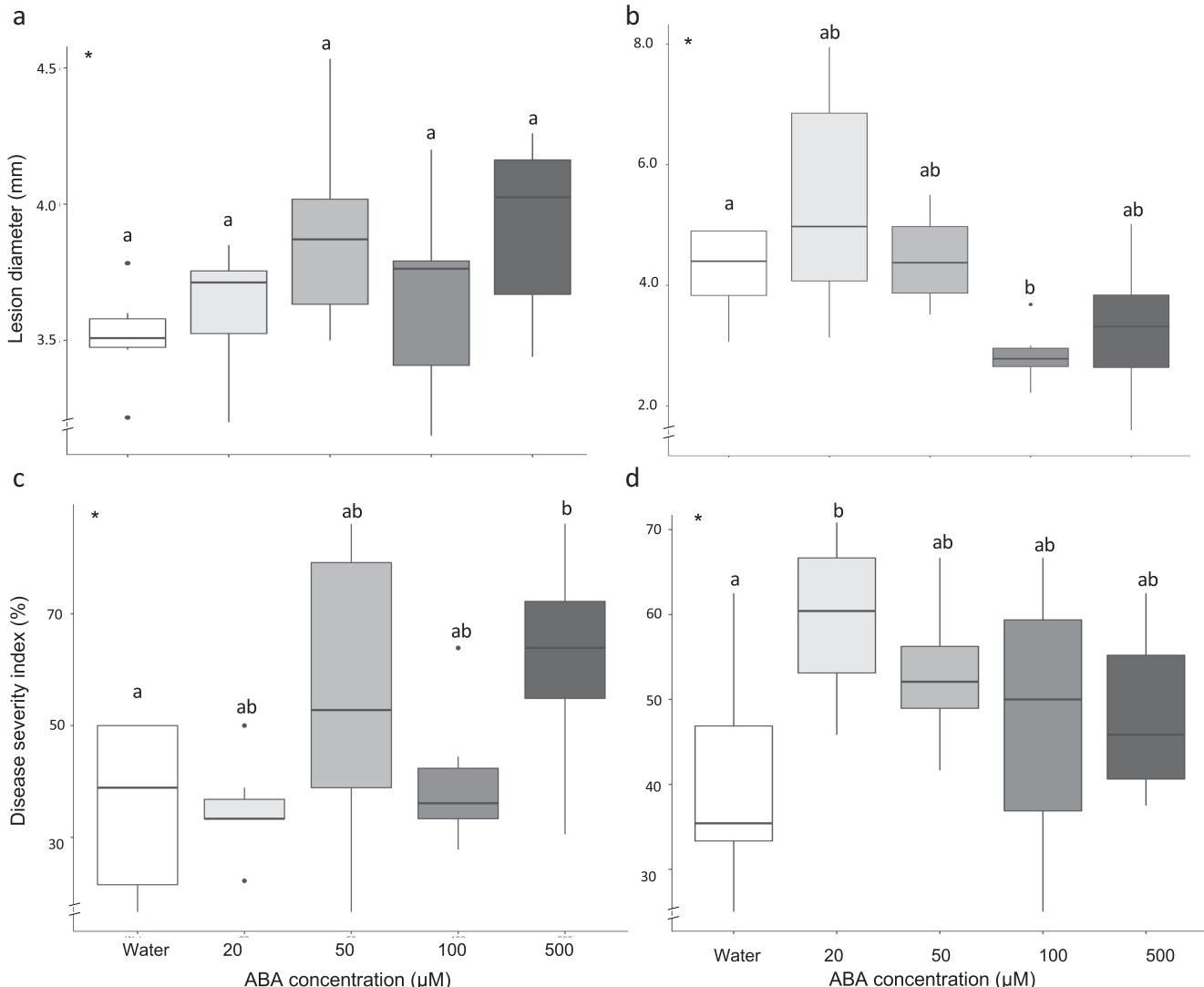

**Fig. 3.** Disease resistance phenotypes in tomato. Four and 8-week-old tomato leaves were sprayed with 20, 50, 100, and 500 $\mu$M of ABA 24 hours pre-inoculation. (a) Lesion diameters in 4-week-old tomato leaves caused by *Botrytis cinerea* at 3 dpi. (b) Lesion diameters in 8-week-old tomato leaves caused by *B. cinerea* at 3 dpi. (c) Disease severity index in 4-week-old tomato leaves converted from the percentage of lesions in six disease categories at 7dpi with *Phytophthora infestans*. (d) Disease severity index in 8-week-old tomato leaves converted from the percentage of lesions in 6 disease categories at 7dpi with *P. infestans*. Boxes denote 25th and 75th percentile, bars display min and max values. Asterisks indicate statistically significant differences at each specific age and treatment (ANOVA for a, c and d, Kruskal–Wallis for b; $p < .05$; $n = 6$). Different letters denote significant differences among treatment groups (LSD for a and d and Dunnets' for b and c post hoc tests; $p < .05$; $n = 6$).

resistance against biotrophic pathogens with low exogenous ABA concentrations. When it comes to the effect against fungal pathogens, DT predicts that ABA affects the resistance phenotype in an age-dependent manner. Our DT predicts susceptibility in older plants; however, experimental work has confirmed that ABA triggers resistance to the necrotrophic fungal pathogen *B. cinerea* only in older leaves.

The reasons behind the enhanced resistance observed after ABA treatments in older plants may be due to age-related resistance (ARR). It is well reported that plant age influences pathogen resistance, with young juvenile leaves displaying enhanced susceptibility (Kus et al., 2002). This is often due to important defence functions, such as PR gene expression, being upregulated in older leaves (Kus et al., 2002; Li et al., 2020). Moreover, different developmental stages, such as the transition to floral stage, are associated with increased resistance in a range of species including Arabidopsis, tobacco and tomato (Develey-Rivière & Galiana, 2007; Hu & Yang,

2019). This resistance has been linked to a positive correlation between development and defence-related processes such as phytoalexin production and resistance gene expression (Bell, 1969; Li et al., 2020). The DT model produced in this study however did not predict this ARR. This is likely due to the small age interval of our data, as 3–6 weeks seems to be the most popular age range for infection studies (Figure 1, Supplementary Table S1). Furthermore, a limitation of our data collection could also be responsible for the lack of ARR in the DT predictions, mostly based on a threshold of age. Whereas we have recorded and assumed that below ~4 weeks old is considered 'young' and ~8 weeks is considered 'old', this may not be true for all plant species studied. For instance, a 4-week-old Arabidopsis plant may no longer be considered young, and an 8-week-old orange tree cannot be considered old. A more accurate assessment of ARR would therefore come from the analysis of the impact of ABA in resistance at different developmental stages for each plant species.

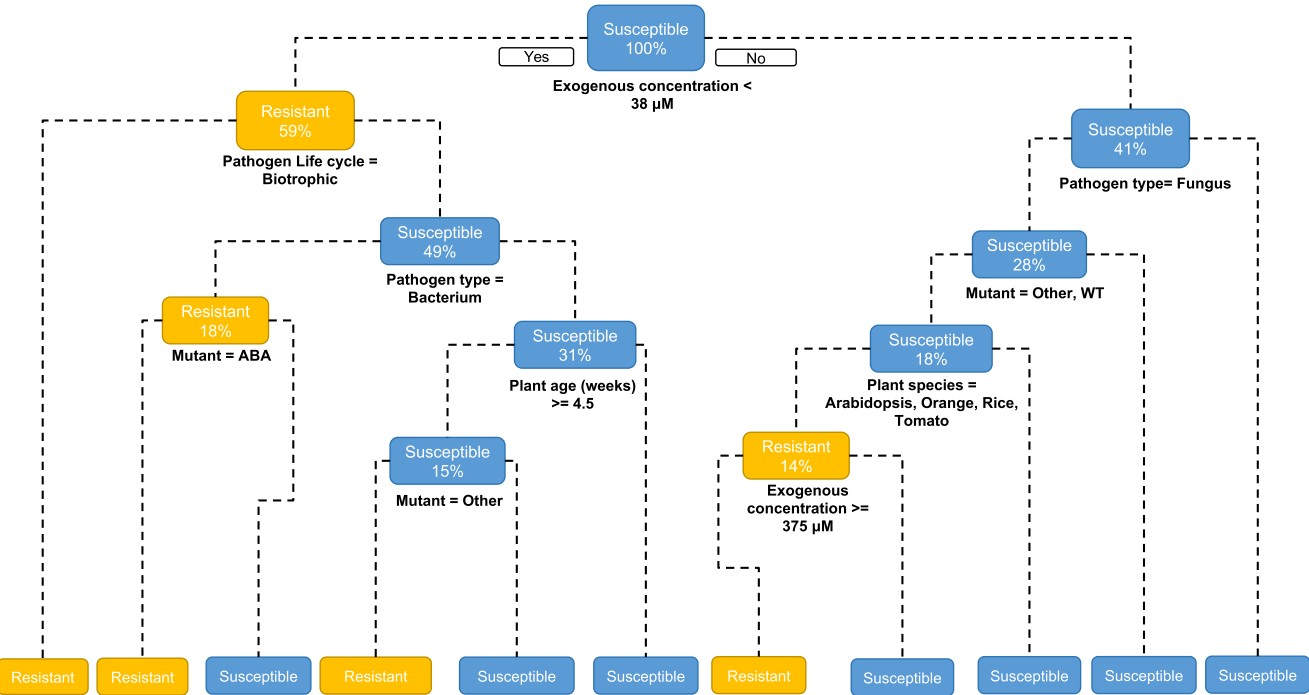

**Fig. 4.** Decision tree (DT) including our experimental data predicting a binary resistance response of 'Susceptible' or 'Resistant'. Yellow boxes indicate resistance and blue boxes indicate susceptibility. Percentage number in box represents proportion of dataset remaining following node split. DT is read from top root node to bottom leaf nodes, if the question asked at a node is 'yes', the left side is followed. Data are described in Supplementary Table S2.

Our results show that pathogen lifestyle determines the effect of ABA in resistance; this can be linked to the modulation of major defence networks associated with the infection strategies. It is well characterised that whereas biotrophic pathogens result in the activation of SA-dependent defences (Mohr & Cahill, 2007), the plant activates JA and ET-dependent defences in response to necrotrophic pathogens (Mcdowell & Dangl, 2000). Moreover, there is a comprehensively understood antagonistic interaction between SA and JA (Anderson et al., 2004, De Torres Zabala et al., 2009). It is known that *B. cinerea* produces a virulence factor, EPS, that directly modulates the JA/SA antagonism. EPS suppresses JA defences, reduces *proteinase inhibitor I* and *II* (*PI I* and *PI II*) expression and leads to the accumulation of SA and NPR1 expression (El Oirdi et al., 2011). Tomato hosts resistance to *P. infestans* can occur through production of defence proteins that have been shown to be ET and SA dependent (Smart et al., 2003). Importantly, it is also known that there are multiple antagonistic responses between ABA and SA, JA, and ET. Therefore, it is easy to speculate that the effect of ABA in resistance against pathogens with different lifestyles is the result of ABA impacting the activation of those hormone-dependent signalling pathways. To the best of our knowledge, this is the first example of the impact of exogenous ABA application on tomato resistance against the biotroph *P. infestans*. Previous studies have been carried out on potato slices against this pathogen demonstrating that ABA treatments increase susceptibility (Henfling et al., 1980; Liu et al., 2020). Our experimental work also demonstrates that treatments with 500 $\mu$M ABA result in susceptibility against *P. infestans* (Figure 3c). Therefore, it is likely that this phenotype is due to disruption of SA defence signalling by ABA. Future studies analysing both genotypes altered in SA and JA signalling will help confirm if ABA application is contributing to susceptibility due to the direct activation or repression of these essential defence pathways.

### 4.3. Application of machine learning tools to complex biological mechanisms

We suggest that tools from ML (here particularly DT and RF classification approaches) provide an approach for further clarification of the role of ABA. This approach is, unsurprisingly, limited by the amount of data available—-as demonstrated by our original DT, which had reasonable predictive performance on test data but failed to capture behaviour in several new experiments, and the reduced performance of our minimised DTs and RFs. For instance, DT predicts resistance in 4-week-old leaves against *B. cinerea* (Figure 2); however, susceptibility is observed experimentally (Figure 3). Additional analyses were conducted to assess how robust our models were with respect to outliers. This demonstrated that extreme ABA concentrations do not dramatically change the structure or predictive capacity of our DT (Supplementary Figure S2). Removing the most extreme age values had a greater structural effect on our DTs: whilst the root node remained as 'exogenous ABA concentration' the value increased to 90 $\mu$M. This indicates that the threshold for susceptibility is dependent on plant age and or tissue type. Later nodes were altered more substantially, though this is not unexpected as an age split appears in the DTs trained on the full dataset, already indicating that higher ages lead to different behaviour. As this outlier-removed analysis removes experiments from fruit material, it is easy to speculate that the observation on the change in ABA concentration may be a tissue-specific effect related to the known tight link between ABA and fruit development and ripening.

These approaches are not proposed as an immediate solution, but rather as a framework for iteratively refining and improving our understanding. Hence, in an attempt to improve the prediction of the initial data we added our experimental data to the model (Figure 4). This improved our DT predictions in the

tomato-*B. cinerea* pathosystem, increasing correct predictions from 1/8 to 6/8. Therefore, the addition of only 16 new data points (i.e., 7.1% change in the original data) led to changes in DT predictions and far greater accuracy against our own experimental results. This improvement points to the importance of adding under-represented data to the model. Here, the capacity for improvement can likely be attributed to the limited current dataset, in particular the narrow age range represented in tomatoes with no data points included in leaves from plants older than 5.5 weeks. We believe that these ML approaches can both help unify and interpret observations from heterogeneous studies and to identify relevant characteristics in the role of ABA in defence. As more data are added, particularly on sparsely represented cases and surrounding key tree-splitting points, further clarity and prediction refinement will be achieved. Thus, we invite researchers to input their results into the model, especially from data in under-represented groups, to improve its prediction capacity. This could allow production of refined species-specific predictive models to determine the threshold level of ABA concentration and test specific questions such as if our findings of the age-dependent effect of ABA against necrotrophs is specific to tomatoes or can be extrapolated to other plant species.

### 4.4. Moving forward with experimental work on the role of ABA in defence

Due to its essential role in abiotic defences, ABA signalling is a common target for engineering stress tolerant plants. Therefore, greater understanding of its controversial role in biotic defences is essential to ensure phenotypes are resistant to both abiotic stress and pathogen challenge. Methodologies used in ABA research vary significantly and this is testament to the many functions of ABA and physiological processes in which it is involved. For example, data examined from literature in this study included repeated ABA spray application, root soaking, biosynthesis mutants, signalling mutants and salt stress treatment, all of which will lead to variable contributions to resistance (Achuo et al., 2006; Audenaert et al., 2002; Song et al., 2011). Despite these, we have been able to identify key characteristics that could determine phenotypes. Whereas it is clear that ABA triggers susceptibility against biotrophic pathogens in a concentration-dependent manner, this is not as straightforward against necrotrophic pathogens, where plant age and specific concentrations are relevant. It is important to understand the function of hormone crosstalk in evolution and survival, as they are an essential characteristic to prevent singular resource allocation against a single stressor (Vos et al., 2015). ABA has been demonstrated to be fully involved in the expression of priming of defence, which does not result in the direct activation of defence mechanisms but the fine-tuning of the response upon subsequent challenges. Due to the key role of ABA in defence crosstalk and priming, it could be hypothesised that ABA plays a central role in buffering any single effort, contributing to noisiness, and therefore playing a central role in the evolution of defence strategies. Therefore, this study can serve as a tool to test that hypothesis and to guide future studies involving the role of ABA in disease resistance.

### Acknowledgements

The authors thank Ana Pineda for her support with the first draft of this paper through the scientific writing online course 'I focus and write'.

**Financial support.** This work was funded by the BBSRC Future Leader Fellowship BB/P00556X/2 to EL and the Midlands Integrative Biosciences Training Partnership (MIBTP) iCASE studentship to KS. EL and KS also acknowledge the pump-priming funding received by the Horticultural Quality and Food Loss Network (WXA3189N/P16188/UoB_Luna-Diez) which has allowed part of this work.

**Conflicts of interest.** The authors declare no conflict of interest.

**Authorship contributions.** E.L. conceived and designed the study and obtained core funding. K.S. conducted data gathering. K.S. and I.J. designed the data analysis pipeline. K.S. performed statistical analyses with strong guidance from I.J. K.S. and E.L. performed laboratory experiments. K.S. and E.L. wrote the article with input from I.J.

**Data availability statement.** All data and code are publicly available at https://github.com/PlantPriming/ABAresistance.

**Supplementary materials.** To view supplementary material for this article, please visit http://doi.org/10.1017/qpb.2023.1.

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
