## [Reviewer Report]

26th May, 2022

Dear editor,

We are delighted to submit our manuscript entitled “Data science approaches provide a roadmap to understanding the role of abscisic acid in defence” to Quantitative Plant Biology. Our manuscript describes a roadmap towards addressing the controversial role of the plant hormone abscisic acid in defence responses against pathogens. We believe that this paper will catch considerable attention in the field of Plant Pathology and will provide a tool for collective effort.

Sincerely,

Dr Luna-Diez and Ms Stevens

---

## [Reviewer Report]

*Comments to Author*: Identifying the role of ABA in the defense against abiotic and biotic stresses in plants is a complex topic to comprehend due to the many factors involved in its action. Several experimental studies exist, but somewhat controversial reports can be found in the literature. In order to tackle this complexity of ABA in plant defense, the authors utilized a computational approach, which the reviewer thinks is innovative in that it provides not only a clue to interpreting the previous study results in the literature but also the future research directions of the subject.

One major question is how the authors justified the use of the DL and the RF models for different plant species. Different species would have different concentrations of ABA in action and specificities to the pathogens used in the individual studies. Concentration, species, and pathogen lifestyle would have tight relationships. As such, wouldn’t it be more reasonable to build models for each species and pinpoint the threshold level of ABA concentration species by species?

Other minor concerns are as follows:

Ln 68: Add the stomatal development with a reference.

Ln 110: machine learning (ML)

Ln 129: Could it be explained in a more detailed way? Keywords for the search, criteria to select the 30 papers, and the methods used in data extraction can also be provided.

Ln 154: random forest (RF)

Ln 162: the package called rpart (o)

Ln 193: Instead of PPFD, report daily light integral (DLI) with the unit of “mol m-2 d-1”.

Ln 206: “cultivation” is not described in the text.

Ln 237: “Experiments were repeated twice with similar results”. This sentence needs clarification. Does this mean the second experiment produced similar results to the first one and the two datasets were used to retrain the models?

Ln 253: represented (o)

Ln 254: ranged (o)

Ln 322: ?50uM. Delete this.

Ln 347: Isn’t it A and B? And C and D?

Ln 353-359: Perhaps, these sentences are discussion material, so I suggest that these should be relocated to the Discussion section. Or to the Materials and Methods section to add the justification for the DT2.

Ln 392-393: Convert these concentrations to the ‘uM’ unit and report them next to their existing unit (in parentheses?).

Ln 483: Delete “Importantly”.

Figure 1G: Add the unit of the concentration. Also, explain why there were 100+ counts for the 0 concentration in the text.

---

## [Reviewer Report]

*Comments to Author*: This article made an interesting job of applying DT and RF in analyzing the plants’ susceptibility to different variables, including plant species, age, ABA concentration, and pathogen. The experimental validation is also impressive.

The most fundamental concern is: How is the label “resistance/susceptible” decided? And Why?

In this paper, Percentage resistance change is used to decide label: positive for resistant, negative for susceptible. If ‘Percentage resistance change’ is calculated as:

Line 145 ‘Percentage resistance change’ = 100 - (ABA value*100/Control value)= x%

Can it be regarded as the binary label is calculated from ABA level change? Are the authors using ABA to predict a label calculated from ABA?

Why more reliant phenotypic features such as lesion area/percentage is not used to decide the label?

Some other questions:

1)What is the precision/recall in the DT/RF model? How much better than a random guess? Why only AIC was shown in article, not ROC/AUC?

2)The resistance of plant to pathogen is not 1/0, why you used binary classification? Is it because the dataset is too small, so it is the current best practice?

3)The importance of some features, such as tissue is not included in table 1. Is it because it is too trivial?

4)There are 14 data points is tissue “fruit” than “leaves”, 12 out of 14 are “susceptible”. However, the leaves sample is more balanced, 94:132=Resistance : susceptible, how do authors interpret this?

---

## [Reviewer Report]

*Comments to Author*: Dear Luna-Diez Estrella and colleagues,

Thank you for choosing QPB for presenting your work and for your patience for the delayed response from our side. We indeed had encountered problems in identifying suitable reviewers who were willing to review your work during the summer and afterwards, and finally secured two review comments recently. As a handling editor, I apologize for the delay and inconvenience that it might have caused.

Shortage of reviewer pool reflects the fact that stress biology field is at the awakening stage of realizing the importance of applying machine learning tools. As you articulated throughout the manuscript, the field lacks a consensus with difficulties in integrating data/pathways that are seemingly under discrepancies.. Two reviewers and I appreciate the authors’ efforts and ingenious approaches to put this work in QPB, and I would like to share my sincere thanks to the authors for the accomplishments evident in this work. We found that the strength of the work lies in the thorough curation of existing data to build the testable model as well as in the follow-up refinement of the model through the experimental validation. The input of small-sized experimental data indeed made a large impact to generate the next model (DT2), which is an attractive demonstration of the utility of ML for the future experimental design in a general framework. As nicely discussed in discussion, this attempt showcases that ML could help experimentalists in finding out the nodes that would result in drastic changes to develop a further experiment to probe for mechanistic explanations. Given that the stress biology field generally suffer from the lack of clues in integrating known pathways, this work can be instrumental when applied to a defined problem set. At the same time, the reviewers and I identified several weak links that would require further investigation and answers.

As this work will set the stage at QPB to showcase ML tools guiding the respective biologists for establishing a testable model to refine their biological questions, I would like to suggest the following revision points. Please understand that our proposed revision points will be focused on making the work translatable to biologists in a more efficient way.

With my own assessment and two reviewers’ comments, I propose the following five major revision points, followed by minor typographic points to fix. Additionally, you shall address other non-redundant points listed in individual reviewers’ comments. As all major comments are included in the five points below, you use your discretion to select non-redundant ones and provide answers in point-to-point form in the rebuttal letter.

Again, I hope that the authors could take this opportunity to improve the manuscript for better communication with experimentalists so that a guided design could emerge to solve a problem. Should you have any further questions and inquiries regarding revision, please e-mail QPB and me (dbsce@nus.edu.sg) for further communication. Direct communication from this point on will surely speed up revision process.

Best regards,

Eunyoung Chae

Major Points

1.We highly recommend the authors to include strong justification of incorporating the results from multiple species across different taxa, and furthermore perform the analyses with a refined dataset (related to major comment from Reviewer 1).

From the validation, the authors have demonstrated current DT is strongly affected by the addition of small dataset. In other words, this can be interpreted as the topology of tree is not robust due to perhaps noises from extremely non-fitting dataset. Stress responses, especially responses to pathogens, underwent extensive diversifying selection to cater specialist pathogens, not to mention the endogenous ABA concentration and tolerance to the exogenous application. The authors are encouraged to perform additional ML analyses with a revised input dataset after subtracting the data from species showing the responses at the extreme ends and/or treatment disparities (e.g. curation of ABA concentration: exo- vs. endogenous). This can be carried out iteratively to refine the performance to the level of cross validating the experimental refinement.

2.Clarify the way the DT is generated and elaborate the DT comparison.

Line 273-278 and figure legend should be improved to further elaborate the details. Legend should include general guideline to interpret and follow the decision-making processes, which totally follows hierarchical flow of positioning important parameters. One could make intuitive guesses, but more explanation is needed. For example, clarify what the percentage in a box means, how one may follow the bifurcation in the tree, and the direction of decision (yes or no: such that ‘with the matching = sign, the decision follows to the deepest drop’).

DT1/DT2 comparison shall be also highlighted with the topology differences: Does the additional appearance of boxes in the bottom row mean the refinement in gaining a prediction power for defining resistance/susceptibility? When it comes to DT1/DT2 comparison, how would the authors explain the shift of ratio of blue/orange boxes?

3.Methods of normalizing heterogeneous values of disease and ABA concentration, and moreover the relation between the two shall be clarified.

Apparently, the current descript is confusing (see Review 2’s comment), while the description in the method section holds true when carefully being read. DSI shall be introduced first and the authors may revise the term “Percentage resistance change” in Line 145 as the function itself only has ABA values as the main parameter.

4.ABA concentration, for example indicated in the Table and DT, shall be clarified as a concentration of either exogenous application or endogenous concentration. If input data has both, this heterogeneity might be the point that shall not be normalized.

5.Figure legend and text indicators for Figure 3 do not match.

There are only four panels in the presented Figure, while the text pointing Figure 3A;C;E;G. In addition, the result of Figure 3A did not seem to be explained with text description; it appears that there are no obvious changes in the figure (Line 319-321). Generally, this result needs a better description to correctly interpret the data.

Minor comments

Line 100-103: What is the major difference between the first and second examples of Pseudomonas treatment? I bet the authors wanted to indicate differences in the usage of wild-type vs. ABA mutant with the treatment of same strain of Pseudomonas syringae (DC3000). In this case, this sentence needs a revision to clarify/specify pathogen strain and keep the consistency on the usage of the strain name within the sentence.

Line 108: remains to be understood.

Line 110: progress on our understanding of such complex questions,

Line 120: revise “affect the effect”, which is seemingly redundant in word choice.

Line 196: freshly prepared stock of ABA

Line 197: Please indicate the stock concentration made in ethanol, which was used to make a dilution series.

Line 198: The control treatment was made by applying water.

Line 200: remove “s” after week. “4-week old plants” is correct, not “4-weeks old plants”.

Line 200: lying flat to laid flat

Line 207-208: This sentence is grammatically incorrect with two major verbs. Please split or revise.

Line 432-434: This sentence is missing a verb.

Line 432: punctuation mark spacing to fix.

---

## [Reviewer Report]

23rd November, 2022

Dear editor,

We are delighted to resubmit our manuscript entitled “Data science approaches provide a roadmap to understanding the role of abscisic acid in defence” to Quantitative Plant Biology. We appreciate the work of the reviewers and yourself to put all the comments together. All comments have been addressed.

Sincerely,

Dr Luna-Diez

---

## [Reviewer Report]

*Comments to Author*: The reviewer has acknowledged that the authors have successfully tackled all of the concerns raised by the reviewers.

---

## [Reviewer Report]

*Comments to Author*: In review of last draft, I’ve already pointed out that, Percentage resistance change is calculated from ABAphenotype. The DSI and percentage resistance change need more explanation.

This paper used a decision tree to study the importance of ABA in disease resitance. the novelty of amalgamated data and explainabilibty of ML model is amazing.

---

## [Reviewer Report]

*Comments to Author*: Dear Luna-Diez Estrella and colleagues,

Thank you for turning in the thoroughly revised manuscript to us. All the reviewers and I appreciate the efforts you made to address major issues. The current manuscript stands nicely to demonstrate the utility of ML approaches to disentangle a complex biological problem and build a testable hypothesis. This notion is well articulated especially in the discussion with a fair comparison of DT1 and DT2 (page 21-22). To compensate the delay encountered during the peer-review process, I would like to suggest a proceeding to publication without further invitation of revision. Yet, there are several minor points that you may need to address in the submission of final manuscript.

Please note that Reviewer 2 still suggested a better description on DSI and percentage resistance change values. I appreciate the changes made in R1 in Line143-148, but I also wondered if there can be an articulation of this value. For example, it would be easy to follow if the authors could give out an example of the % value when an experiment gives rise to an obvious change in resistance in an ABA-dependent manner.

Other typographic errors are listed below, which can be handled during a final check and submission from your side.

Page 6: Is the parenthesis in the “(maximum likelihood)” necessary?

Line 222-223: Add “and” before “, plates were kept..”.

Line 271-272, 308 and elsewhere: fix uM to the proper greek letter form.

Line 282: Does “Exogenous” need to be capitalized?

Line 534-536: This sentence needs revision, possibly adding “and” after “,”.

Line 542: : Fix the “.,” and start a new sentence with “It”.

---

## [Reviewer Report]

Dear editor,

Many thanks for the revision of our paper. We are delighted that this manuscript has been accepted for publication. We have done the final amendments in the paper as indicated by the reviewers and yourself.

We also changed the graphical abstract to make it more engaging.

Best wishes,

---

## [Reviewer Report]

*Comments to Author*: Dear Estrella Luna-Diez,

Thank you for submitting the finalized revision to us. I hope that our peer review process was useful for you and your coauthors to make your exciting work to be shared with quantitative plant biology readers. Again, my apologies to the delayed review processes. As an editor, I look forward to seeing this work guiding data scientists and experimentalists to put forward a new avenue to follow for stress biology and hormonal controls.

Best,

Eunyoung